# A Combinatorial *Q-Locus* and Tubulin-Based Polymorphism (TBP) Approach Helps in Discriminating *Triticum* Species

**DOI:** 10.3390/genes13040633

**Published:** 2022-04-01

**Authors:** Chiara Guadalupi, Luca Braglia, Floriana Gavazzi, Laura Morello, Diego Breviario

**Affiliations:** Istituto Biologia e Biotecnologia Agraria, Via Edoardo Bassini 15, 20131 Milano, Italy; c.guadalupi@outlook.com (C.G.); luca.braglia@ibba.cnr.it (L.B.); floriana.gavazzi@ibba.cnr.it (F.G.); laura.morello@ibba.cnr.it (L.M.)

**Keywords:** *Q-locus*, TBP, spelt wheat, *Triticum*, food authentication

## Abstract

The simple and straightforward recognition of *Triticum* species is not an easy task due to their complex genetic origins. To provide a recommendation, we have compared the performance of different PCR-based methods relying on the discrimination ability of the *Q-* and *γ-gliadin* (*GAG56D*) genes, as well as TBP (Tubulin-Based Polymorphism), a method based on the multiple amplification of genes of the β-tubulin family. Among these approaches, the PCR-RFLP (Restriction Fragment Length Polymorphism) assay based on a single-nucleotide polymorphism (SNP) present in the *Q* gene is the only one capable of fully discerning hexaploid spelt and common wheat species, while both *γ-gliadin* and TBP fail with similar error frequencies. The *Q-locus* assay results in the attainment of either a single fragment or a doublet, depending on the presence of a suitable restriction site, which is affected by the mutation. This dual pattern of resolution limits both the diagnostic effectiveness, when additional *Triticum* species are assayed and compared to each other, and its usefulness, when commercially available flours are analyzed. These limitations are overtaken by flanking the *Q-locus* assay with the TBP analysis. In this way, almost all of the *Triticum* species can be accurately identified.

## 1. Introduction

Wheat is the second most cultivated cereal worldwide after maize. In total, 95% of the global wheat production, which amounts to 760 million tons (USDA World Wheat Production 2020/2021 Circular Series WAP 7–21 July 2021), is made up of *Triticum aestivum* L. ssp. *aestivum*, the hexaploid cultivated species usually called bread or common wheat [1]. The remaining 5% is substantially made up of *Triticum turgidum* ssp. *durum* (durum wheat). Among the hexaploid series of the *Triticum* genome (AABBDD), spelt wheat (*Triticum aestivum* L ssp. *spelta* (L.) Thell.) has gained a growing interest because of its great adaptation to a wide range of environmental conditions, depending on certain additional agronomic properties such as the efficient assimilation of nitrogen, the excellent growth capacity in poor soil and the high disease resistance. Altogether, these features make spelt wheat particularly suitable for breeding programs that aim to develop varieties characterized by high grain quality and high resistance to pathogens [2]. Together with einkorn (*Triticum monococcum* L.), emmer (*Triticum turgidum* ssp. *dicoccum* L.) and Khorasan (*Triticum turgidum* L. ssp. *turanicum* (Jakubz.) A. Löve and D. Löve), spelt wheat defines a group of ancient wheat species, which capable of growing under low input and organic farming, have attracted interest in the grain market, gaining a price 25 per cent higher than that of common wheat, although the claim for their superior nutritional features remains disputable [1,3].

During the long domestication process, which has led to the currently cultivated forms of wheat, two of the agronomical traits have most significantly contributed to the higher yield of the modern varieties—the loss of shattering of the spike at maturity and the change from hulled to free-threshing naked forms [4]. How these traits have been differentially segregating at a whole-genome level among spelt and common wheat remains to be fully deciphered. One possible explanation for the onset of the European spelt is that after its migration to Europe from the Fertile Crescent area, a free-threshing hexaploid wheat hybridized to a hulled tetraploid emmer wheat. This event eventually translated into the genetic differentiation of the A and B subgenomes in common wheat and European spelt, with no contribution by the D subgenome [5].

Once this differentiation took over, spelt was cultivated in Europe from the bronze age until recent times, when free-threshing common wheat gradually replaced spelt cultivation because of its grain characteristics being much more suitable for mechanical harvesting and seed processing. This led to a strong decrease in spelt cultivation in Europe in the 20th century, limiting it to small areas of a few European regions [5,6]. However, the two *T. aestivum* hexaploid subspecies, common wheat and spelt, can be freely intercrossed, a strategy that breeders continue to exploit to transfer agronomically important genes from spelt into the common wheat gene pool, with the aim of generating new varieties [2,5]. Breeding can, therefore, lead to the production of a couple of spelt types, namely “pure spelt”, resulting from intercrossing local spelt populations, and “crossed spelt”, obtained via the hybridization of spelt with common wheat varieties. The increased use of alternative wheat species in response to both process and market demands has boosted the urgent need from industrial millers and bakers for a rapid, effective and inexpensive discrimination method for kernels and flours.

As occurs in many lines of investigation concerning species authentication in raw food materials and mixtures, DNA-based methods have eventually taken the lead over other classical biochemical methods due to their handiness, convenience, specificity and rapidity of execution. In fact, spelt and common wheat could in principle be discriminated by measuring the lipid content and composition [7], although the procedure is expensive, labor-intensive and time-consuming, making it unaffordable for the cereal industry. Additionally, infrared-based methods such as near-infrared spectroscopy (NIRS) and attenuated total reflection–Fourier transform infrared (ATR–FTIR) spectroscopy could be successfully used to differentiate hexaploid *Triticum* species on the basis of their protein amount and composition [8,9,10], but these techniques can be easily affected by many environmental factors and agronomical treatments, thereby leading to possible misclassifications [11,12].

Among the DNA target sequences shown to be very effective in discriminating spelt from common wheat, the *Q-locus*-based method has gained prominence [13]. This is not accidental, since the Q gene is a key determinant of spike morphology, thereby influencing many important agronomical traits [14]. In fact, located on the chromosome 5 of the A subgenome, the Q gene encodes for a transcriptional factor of the APETALA2 (AP2)-like family involved in the determination of the rachis fragility, glume shape and tenacity and spike length. Its expression is regulated by both miRNA172 accumulation and TOPLESS co-repressor activity [15]. Two functional alleles, Q-5A and q-5A, are differentially distributed in common wheat and spelt, respectively [16]. The dominant Q allele is associated with high levels of transcript and a more compact spike morphology of a free-threshing grain, while the recessive q-5A allele, present in the European spelt, is associated with the hulled phenotype. Interestingly, certain Asian spelt accessions have been reported to carry the Q allele found in common wheat, which has led to the hypothesis that the expression level of the Q-gene and the control of the spike morphology ultimately depend on the genetic background [5]. The Q and q alleles in the A subgenomes of hexaploids differ in their six conserved single-nucleotide polymorphisms (SNPs) [14]. In particular, a G to C transition within exon 8 close to the AP2 domain regions results in non-synonymous substitution from valine to isoleucine, while a neutral C to T substitution occurs within exon 10 at the miRNA172 binding site [17]. These SNPs have been used to develop different PCR-based approaches to discriminate spelt from common wheat. In addition, full-length sequencing of the Q-5A gene revealed a unique deletion in the 5Aq allele present in some European spelt germplasms, suggesting a direct inheritance from the tetraploid ancestor *T. turgidum* ssp. *dicoccum* [17]. Koppel at al. [3] have recently developed a duplex droplet digital PCR (ddPCR) method, which targeting the exon10 C/T polymorphism, allows the detection and quantification of contamination by common wheat in food products made from spelt. Similarly, Morcia et al. [13], exploiting the same mutation, have recently developed a chip-based dPCR method for wider discrimination of hulled versus hulless wheats and for the relative quantification of their percentage amounts in flour and flour-based products. Since the same q allele is also present in other commercial hulled wheats, all of these methods can discriminate between hulled and naked wheats but not between tetraploid, exaploid and diploid species, e.g., spelt from emmer and einkorn.

The *γ-gliadin*-encoding locus, located on subgenome D (*GAG56D*), has also been used to discriminate between spelt and common wheat by exploiting a couple of polymorphic traits that are contributed by either an SNP (A/G) or a tandemly repeated nonamer (CAAGAACAA). In common wheat, the latter defines an insertion in one of the conserved regions of the C-terminal domain (Von Buren et al., 2000). However, based on recent studies reporting the occurrence of different rearrangements on the D subgenome, the 9 base pair repeat insertion can no more be confidently assigned to common wheat. In fact, the largely uncontrolled process of evolution from spelt to common wheat in some cases has meant that spelt has acquired the nonamer repeat, while some others common wheat varieties have lost it [11].

The TBP (Tubulin-Based Polymorphism) method, based on the presence of ubiquitous yet variable plant β-tubulin loci, was instead reported by Silletti et al. [18] as a convenient DNA fingerprinting tool for the genetic identification of most common food cereals and commercialized species belonging to the *Triticum* genus, which is achievable independently of their hulled or hulless seed phenotype. Based on limited evidence, authors have suggested that TBP could also distinguish spelt from common wheat, depending on the presence of an additional 581 bp fragment in the amplification profile of the latter. Here, we further investigated this possibility by analyzing 14 spelt and 22 common wheat varieties and comparing the TBP data with those obtained with the *Q-locus* assay as the gold standard, as well as with the *GAG65D* assay. This comparison was further extended to the analysis of ten commercialized flour samples derived from different *Triticum* species.

## 2. Materials and Methods

### 2.1. Plant Material and Flour Samples

Seeds from wheat and related species and subspecies considered in the present paper (Table 1) were courteously provided by DISTAL, Department of Agricultural and Food Sciences, Alma Mater Studiorum Università di Bologna. In addition, a panel of 36 spelt and common wheat accessions (seeds), including currently cultivated or ancient cultivars, as well as landraces, was provided by d CREA-AA, Consiglio per la ricerca in agricoltura e l’analisi dell’economia agraria—Agricoltura e Ambiente, Headquarters of Foggia; and CREA-FLC, Consiglio per la ricerca in agricoltura e l’analisi dell’economia agraria—Centro Ricerca Produzioni Foraggere e Lattiero Caseari (Table 1). Regarding spelt accessions, both pure strains (PS) and germplasms crossed with common wheat (CS) were considered, according to the provided pedigree and origin information.

The accessions referring to commonly cultivated cereal species included in the analysis are part of the CNR IBBA germplasm collection.

Cereal-based flour samples were kindly provided by Mirtilla Bio srl bakery (Table 2, samples A–I) or bought online from specialized Italian companies (Table 2, samples L–N).

### 2.2. DNA Extraction

Seed and flour samples were ground to a fine powder (5–10 µm) according to the protocol developed by [18] and 100 mg samples were used for the extraction of the total genomic DNA (gDNA) using the spin-column-based DNeasy Plant Mini Kit (Qiagen, Hilden, Germany) as modified by [19]. The gDNA concentration and purity were determined fluorometrically using the Qubit^®^ dsDNA BR assay kit (Qubit 1.0 fluorometer, Thermo Scientific, Waltham, MA, USA) in accordance with the manufacturer’s instructions and by measuring the UV absorbance ratios at 260 and 280 nm with a micro-volume spectrophotometer (Nanodrop, Thermo Fisher Scientific). The DNA integrity was also evaluated by loading 3 µL of gDNA in a 0.8% agarose gel and using DNA ladders for reference.

### 2.3. Q-Locus and γ-Gliadin-D Assays

The amounts of gDNA used for the amplification of both the Q (*Q-locus*) and *GAG56D* (*γ-gliadin-D*) genes were 50 ng when extracted from seeds and 100 ng from cereal-based flours. The primer pairs were those described by [11] and referred to region 1, introns 1–3 of the *Q-locus,* in accordance with the gene representation provided by [14]. The PCR reactions of both assays were carried out in 25 µL reaction volume, including the 2× VWR Taq Polymerase Master Mix containing 2 mM MgCl_2_ (VWR International, Pennsylvania, USA) and 0.5 µM of each primer. PCR reactions were performed on a Mastercycler ×50 (Eppendorf srl, Milan, Italy) using the following cycling protocols: *γ-gliadin-D*, 94 °C pre-denaturation 3 min; 94 °C 40 s, 60 °C 30 s, 72 °C extension 30 s, 30 cycles, 72 °C final extension 1 min; *Q-locus*, 94 °C pre-denaturation 3 min, 94 °C 40 s, 58 °C 30 s, 72 °C extension 30 s, 30 cycles, and 72 °C final extension 1 min. The subsequent enzymatic cleavage of the *Q-locus* amplicons was performed with *Msp*I (1U) (Thermo Fisher Scientific) on 10 µL of PCR product at 37 °C overnight to a final volume of 25 µL, followed by an inactivation step at 65 °C for 20 min. The PCR products, either cleaved or uncleaved, were separated by running a 2% (*w*/*v*) agarose gel, which was eventually stained with Atlas ClearSight Gold DNA stain (Bioatlas, Tartu, Estonia).

### 2.4. TBP Profiling and Single β-Tubulin Intron Amplification (TUBB7)

Here, 50 ng of gDNA was used as the template for the TBP amplification of seed samples, while 100 ng was the amount used for the analyses of cereal-based commercial flours. The amplifications of both the 1st and 2nd β-tubulin intron regions were performed using degenerated primers and PCR conditions as detailed by [20]. Two independent TBP amplifications of the same gDNA extraction and two different dilutions of each amplification were always performed for each analyzed sample to ensure both the reliability and repeatability of the analysis.

The amplification of a short fragment of the polymorphic intron sequence of a single β-tubulin gene (*TUBB*7) was performed using the following degenerate PCR primers: 7For GACTGCCTCCAGGGTACGTGC-7Rev CCTGRAATCCTGCAGTGARGAAGA. The 5′-end of the forward primer was labeled with the 6-FAM fluorescent dye to allow the detection of the different fragments once separated by capillary electrophoresis. Here, 50 and 100 ng amounts of template (gDNA) extracted by either seed or flour samples were used for the amplification. PCR reactions were carried out in accordance with the “TBP *light*” protocol reported by [18], with a primer annealing step performed at 63 °C for 50 s.

### 2.5. Capillary Electrophoresis (CE) Separation and Data Analysis

The FAM-labeled amplicons resulting from TBP and *TUBB7* amplifications were first checked for their amounts by loading 4 μL aliquots of each PCR product on a 2% agarose gel, which were then diluted in double-distilled water to a various extent (up to 1:10), depending on the signal intensity of the amplicon compared to that of a 1 Kb plus marker used as a reference (Thermo Fisher Scientific). Typically, two microliters of each diluted CE-TBP and CE-*TUBB*7 amplified sample, with the addition of an appropriate volume of a 1200 or 500 LIZ Size Standard, respectively (Thermo Fisher Scientific), was loaded on the 3500 Genetic Analyzer for CE separation after denaturation at 95 °C for 5 min. The running protocol and data collection procedure were those reported by [20]. Gene Mapper Software v.6.0 (Thermo Fisher Scientific, Waltham, MA, USA) was used to analyze the fluorescence data, assigning the peak size (allele calling) as a function of the size standard. The resulting data were analyzed according to the peak height threshold value defined by [20], collected by GeneMapper software and stored in a standard text data file for subsequent analysis. The CE-TBP or CE-*TUBB*7 numerical data referring to both the size (in base pairs) and the height (in relative florescence units—RFUs) of the peaks resolved in each analyzed sample were converted into corresponding Microsoft Office Excel files. This allowed their alignment according to length, thereby assisting in sample profile comparisons. The CE-TBP profiles resulting from both the 1st and 2nd intron amplifications were converted into binary matrices (1 for the presence and 0 for the absence of a peak) and a neighbor joining tree was inferred from the genetic similarity estimated among genotypes according to Jaccard’s index for binary data, using the open-source software package Past v.4.07b (last accessed on 20 January 2022) [21].

## 3. Results

The assay targeting an SNP in the *Q-locus* (region 1, introns 1–3) of the A subgenome of *Triticum* spp. (Table 1), based on a recently reported PCR-RFPL technique [14], was applied to different wheat species, their ancestors and other cereals. As expected (Figure 1), only *Triticum* species containing the subgenome A showed successful PCR amplification. In addition, the hulless and hulled species could be respectively identified by the presence of either an uncleaved 323-bp-long fragment or a doublet, resulting from the combination of 186- and 137-bp-long fragments, respectively. This different output depends on the presence in the *Q-locus* sequence of a C or T nucleotide at the *Msp*I restriction site. Accordingly, spelt and common wheat, both ABD hexaploids, can be easily discernible by the presence of a cleaved or uncleaved band, respectively. A similar output was observed in wheat tetraploids (AB) with emmer showing a doublet, while durum and Khorasan samples showed a single uncleaved amplicon. Thus, by using this assay, common wheat cannot be distinguished from durum wheat or tritordeum—a cross between durum wheat and a diploid wild barley—while einkorn, emmer and spelt all look alike.

Conversely, the TBP assay performed on the same experimental samples assigned specific genomic profiles to almost all of the wheat species and subspecies analyzed (Appendix A), with the spelt sample (accession ‘Rita’) differs from the common wheat sample based on the absence in the 1st intron profile of a peak corresponding to an allelic variant present in the *TUBB7* locus.

This putative-specific discrimination was further tested in succeeding analyses by including several cultivars of common and spelt wheats, as well as crossed lines with different pedigrees. In fact, two types of spelt wheat, “pure” and “crossed”, can be defined as the result of different breeding strategies (Table 1), which in the crossed type might have led to the introgression of portions of the common wheat genome into spelt, impeding subspecies-specific genetic authentication. Landraces and wild wheat accessions were also added to the analysis. Genetic relationships obtained by scoring the 72 TBP markers resulting from the amplification of both the 1st and 2nd β-tubulin intron regions are reported in the cluster analysis shown in Figure 2. Overall, the tree shows a fine separation of the vast majority of the analyzed wheat species and subspecies, which is in accordance with the domestication history of cultivated wheats and supported by interspecific hybridization and allopolyploidization events. Diploid accessions were grouped separately depending on their genome (A, B or D). More precisely, accessions containing the B genome defined a distinct, separate cluster, whereas wheat accessions containing the A genome formed two distinct subgroups, A^m^ or A^u^, thereby separating *T. monococcum* from *T. urartu.* The former is considered a primitive, Neolithic era domesticated wheat form, while the latter corresponds to a wild species [22]. As also shown in Figure 2*, Aegilops tauschii* ssp. *strangulata*, the only species of reference for the D genome, significantly rooted the separation between the tetraploids (AB) and the hexaploids (ABD) subclusters, being the donor of the D subgenome in the latter. Notwithstanding this meaningful classification, the tree clearly documents the absence of a specific spelt clade independent of the breeding history. As shown in Figure 2, accessions of common and spelt wheats are interspersed within different clades. Thus, the original view that TBP could readily distinguish common wheat from spelt wheat due to the presence in the former of an additional 581bp fragment was dropped.

This led to further investigations performed by limiting the TBP analysis to the use of a single β-tubulin gene—*TUBB*7. Fourteen spelt and 22 common wheat samples were analyzed (Table 1) with the use of specifically designed PCR primers. As reported, the amplification of *TUBB*7 led to the production of two fragments of 288 and 301 bp, respectively. The former amplicon (288 bp) that is associated with subgenome D was present in all of the samples analyzed, independently of the species, whereas the latter (301 bp), associated with the A subgenome, was distributed preferentially but not exclusively within the spelt wheat samples. In fact, differential CE-TBP profiles were obtained depending on the presence or absence of the A-subgenome-derived peak, which was present with variable frequency in spelt and common wheat.

These results were compared to those obtained from the PCR-RFLP *Q-locus* assay, which was used as a reference, and those obtained from the *GAG65D* assay, which was performed as described by Curzon et al. [11] (Table 2). At the *Q-locus*, all 14 spelt wheat samples show the presence of a doublet, regardless of their genetic background, whereas a single uncleaved fragment was detected for each of the 22 common wheat samples, indicating the presence of the recessive q allele. When the same samples were analyzed with either the *γ-gliadin-D-* or the *TUBB*7-specific primers, such neat discrimination could not be achieved and the presence of common wheat-specific amplicons of 236 or 301 bp, respectively, could be detected in the spelt wheat accessions, with a corresponding error frequency of 21% (Table 2) for both markers. However, all three pure spelt accessions, ALT1, SPE and FAR111, are correctly identified by the three markers, whereas incorrect assignation was found for some crossed spelt accessions. Some common wheat accessions also showed the presence of the spelt allele for either the *GAG65D* or the *TUBB7* locus.

The general inferences derived from our results indicate that the TBP method, developed on multiple beta-tubulin loci, although failing in assisting in accurate discrimination between spelt (pure and crossed) and common wheats, offers a wider spectrum of detection among the *Triticum* species in comparison to that achievable with the use of the *Q-locus* alone. In addition, as extensively reported in the literature [18,19], the TBP method can efficiently genotype other cereal species that may be present or used in wheat flour and its derived products.

To assess this, different samples of commercially available wheat-based flours that are commonly used in the bakery and pastry industry were first analyzed using the *Q-locus* assay. According to the producers’ claims, they were pure flour samples, each obtained using one of the following species: common and durum wheat, ‘Khorasan’, spelt, einkorn or emmer (A–M samples, Table 3). One multigrain flour was also included in the analysis (N).

The results of the authentication analysis performed on these samples with the use of the *Q-locus* assay are shown in Figure 3. As is noticeable, five out of six flour samples derived from spelt (A–E) show hybrid restriction patterns, whereby the uncleaved fragments coexist with the doublet, while only sample M shows fully digested fragments. The co-existence, of variable intensity, of the undigested single fragments in samples A–E, (Figure 3) could be attributed to the presence of hulless wheat species such as common, durum or Khorasan wheat. In principle, even the cleaved fragment could hide contamination, i.e., spelt flour samples could contain either einkorn or emmer, given the lack of species specificity of the *Q-locus* assay. Similar observations can be made for the analysis of the two F and G einkorn samples. They show a doublet typical of hulled wheat species; therefore, possible contamination from either spelt or emmer cannot be excluded. A similar reasoning, although applied in a reversed way, can be made for Khorasan-derived flour samples. In fact, samples H and I show a single uncleaved fragment, thereby proving the absence of any contaminating hulled species, although the possible presence of common or durum wheat cannot be excluded. Instead, the emmer sample in lane L shows the presence of one or even more contaminants.

Therefore, the *Q-locus* assay is a useful tool, as it is capable of distinguishing spelt and common wheat, or more generally hulled from hulless wheat species, while nonetheless being unable to support any specific assignment if applied in a market context, whereby the more versatile TBP method could instead be of help. In fact, the same flour samples analyzed by TBP profiling (Table 4) all showed additional peaks referable to the presence of contaminations, with the exception of wholemeal spelt flour M, the only sample that looks to be made from pure spelt wheat. Durum wheat and einkorn were identified as the contaminants in spelt flour samples (A and D), while samples B–E are likely to contain trace amount of common wheat, as shown by the presence of the 581 bp amplicon, which is missing in sample M.

In addition, as reported in the same table, the TBP assay uncovered hexaploid wheat contaminants in einkorn (F, G) and emmer (L), with the latter also containing durum wheat, thereby showing a higher sensitivity over a wider spectrum of analysis. Ingredients declared in the multigrain flour were also identified by TBP assay.

Thus, because of their complementary capacity, the *Q-locus* and TBP assays could be conveniently combined to identify any contaminant *Triticum* species in flour and derived products, with the exception of durum and Khorasan, which are not discernable at present.

## 4. Discussion

In accordance with previous data [17], we have shown that a single polymorphism present in the Q gene was distributed in a distinctly different way between common and spelt wheats in the tested samples, making their reciprocal recognition very effective for both pure and crossed spelt cultivars, while being fully linked to the corresponding free-trashing or hulled grain phenotype. The assay is simple, based on the PCR amplification of a fragment 323 bp long, followed by digestion with the *Msp*I enzyme, which can occur only in the presence of the q allele. This tight spelt–q allele association relates to the fact that the Q gene encodes for a transcription factor that influences several traits, including grain threshing. For this reason, very often the spelt character corresponds to the free-threshing phenotype in crossed spelt lines. However, this is not always the case. In fact, in testing different markers for wheat or spelt discrimination, Curzon et al. [11] reported that only 64 out of 77 hexaploid wheat lines classified as spelt by the providers showed a hulled phenotype after threshing, and accordingly carried the q allele. Therefore the authors reclassified these crossed spelt lines as wheat. This tautological reasoning means that in the absence of morphological evidence, commercialized spelt accessions may have the common wheat-dominant Q allele. Moreover, certain Asian spelt accessions have also been reported to carry the Q allele, suggesting that this genetic marker, although highly significant, cannot be used as an absolute discriminant. Since the Q gene is contributed by the A genome complement, its effectiveness is limited to those wheat species (*Aegilops* spp.) that contain it. This explains why it cannot be detected in those species containing only the B or D genome complements, as well as in other cereals. Misrecognition between spelt and common wheat can instead occur when using either the *γ-gliadin*-*D* or the *TUBB7* assays (Table 2). For both, misclassification can originate from two mutual types of errors, whereby either common-wheat-specific fragments are absent in their own genome or are detected in some of the spelt wheat cultivars (Appendix A). These two types of misclassifications can occur at different frequencies in different and unrelated varieties of both species. When using *γ-gliadin-D* as a marker on a total of 36 samples, the misclassification of spelt wheat due to the presence of the common wheat fragment amounted to 21%, whereas common wheat went undetected in 18% of the analyzed samples (Table 2). With *TUBB7,* these percentages amounted to 21% and 9%, respectively. As mentioned, misclassified cultivars were not the same when comparing the two methods. In fact, while cv. Maddalena, Rossella and Giuseppe of spelt wheat showed a common wheat *TUBB7* profile, *γ-gliadin-D* was detected as common wheat of the Rieti, Pietro and Giuseppe cv. On the other hand, Benco, Carosello, Gentil Rosso Aristato and Mieti were correctly recognized as common wheat by *TUBB7* but not by *γ-gliadin-D.* Conversely, common wheat cv. Marzuolo and Palesio were recognized by *γ-gliadin-D* but missed by the TBP assay. These results were the likely consequence of the breeding history of spelt crossed cultivars, showing different degrees of wheat genome introgression and a past breeding history with spelt.

In accordance with previous data [11,17], our results have further shown that the Q-gene-based assay is by far the best available method for discerning spelt from common wheat when applied to kernels. The question remains regarding the applicability of the *Q-locus-*based assay to the recognition of the botanical origin of commercialized wheat flours. It is in this regard that we have actually shown that the Q-gene-based assay can only be used to ascertain the presence of a fragment profile consistent with the presence of either spelt or common wheat (singleton or doublet), without decisively proving their identity, because the same profile can be contributed by other species. To this end, in order to offer a practical tool allowing wheat species recognition in flour and derived market products, we flanked the TBP analysis to the *Q-locus* based assay. Using this dual approach, we have shown that the majority of *Triticum* species can be effectively recognized with the exception of two subspecies, durum and turanicum (Khorasan), which remain challenging and are still unsolved by any molecular markers to our knowledge. Furthermore, the Q-*locus*–TBP combination, could also help in the recognition of flour made from cereals different than wheat, either declared or not, in the mix used to make a large variety of bakery and pastry products.

## Figures and Tables

**Figure 1 genes-13-00633-f001:**
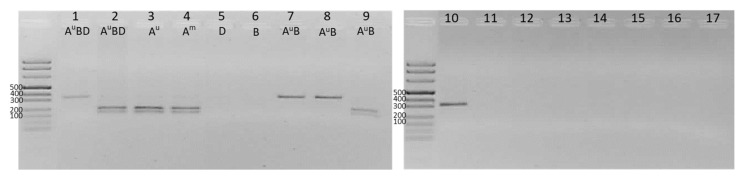
PCR-RFLP analysis of the Q-locus: (**1**) *Triticum aestivum* ssp. *aestivum* ‘Abbondanza’; (**2**) *T. aestivum* ssp. spelta ‘Rita’; (**3**) *T. urartu*; (**4**) *T. monococcum* (einkorn); (**5**) *Aegilops tauschii*; (**6**) *Ae. Speltoides*; (**7**) *T. turgidum* ssp. durum ‘Claudio’; (**8**) *T. turgidum* ssp. *turanicum* (Khorasan); (**9**) *T. turgidum* ssp. *dicoccum* (emmer); (**10**) tritordeum; (**11**) maize; (**12**) rice; (**13**) *sorghum*; (**14**) oat; (**15**) millet; (**16**) rye; (**17**) barley. ExcelBand™ 100 bp DNA Ladder (Smobio) is shown on the left side of both gels.

**Figure 2 genes-13-00633-f002:**
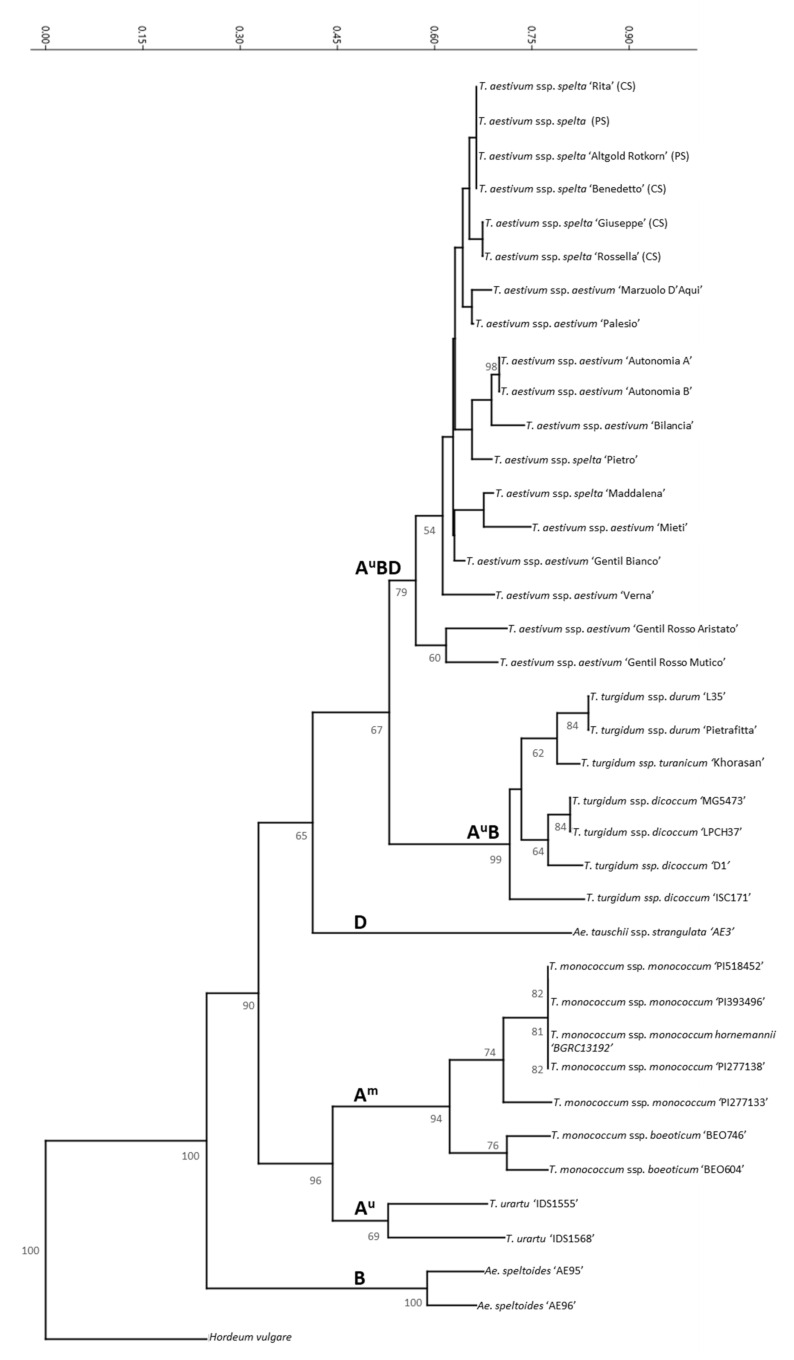
The neighbor-joining tree showing the genetic relationships among *Triticum* and *Aegilops* genera based on TBP analysis (1st and 2nd intron regions). Only bootstrap values higher than 50% are shown. Barley (*Hordeum vulgare*) was used to root the tree.

**Figure 3 genes-13-00633-f003:**
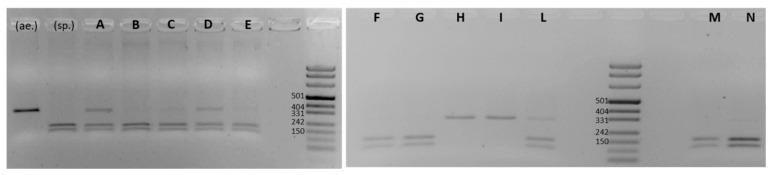
*Q-locus Msp*I-based assay applied to commercial flours obtained from different wheat species. The sample codes (A–N) are from Table 2. On the left side, the profiles of T. *aestivum* ssp. *aestivum* ‘Palesio’ (ae.) and *T. aestivum* ssp. *spelta* ‘Rossella’ (sp.) are shown as references. The pUC8 DNA Marker 8 (Thermo Fisher Scientific) value is shown for each gel.

**Table 1 genes-13-00633-t001:** Different cereal accessions used in the study. The genome type and ploidy level are reported only for wheat related species and subspecies. Spelt and common wheat cultivars and landraces collected from different sources are also included. Concerning spelt, two different breeding types (pure strain or crossed with common wheat) are indicated, and when available the lineage information is reported.

Species	Variety/Landrace/Line	Sample ID	Common Name	Genome	Type	Pedigree	Ploidy	Provider
*Triticum urartu* Thum. ex Gandil.	IDS1555	-	Wild wheat form	A^u^	-	-	2×	DISTAL
	IDS1556	-			-	-		
*Triticum monococcum* L. ssp. *beoticum*	BEO746	-	Wild einkorn wheat	A^m^	-	-	2×	DISTAL
	BEO604	-			-	-		
*Triticum monococcum* L. ssp. *monococcum*	PI518452	-	Einkorn wheat	A^m^	-	-	2×	DISTAL
	PI393496	-			-	-		
	PI277138	-			-	-		
	PI277133	-			-	-		
	hornemannii BGRC13192	-			-	-		
*Aegilops speltoides* ssp. *speltoides* Tausch	AE95	-	Goatgrass	B	-	-	2×	DISTAL
	AE96	-			-	-		
*Aegilops tauschii* ssp. *strangulata* (Eig) Tzvelev	AE3	-	Tausch’s goatgrass	D	-	-	2×	DISTAL
*Triticum turgidum* ssp. *dicoccum* Schrank	MG5473	-	Emmer wheat	A^u^B	-	-	4×	DISTAL
	LPCH37	-			-	-		
	D1	-			-	-		
	ISC171	-			-	-		
*Triticum turgidum* ssp. *turanicum* Jakubz	Khorasan	-	Khorasan wheat	A^u^B	-	-	4×	DISTAL
*Triticum turgidum* ssp. *durum* (Desf.) Husn.	Claudio	-	Durum wheat	A^u^B	-	-	4×	DISTAL
	L35	-			-	-		
	Pietrafitta	-			-	-		
*Triticum turgidum* ssp. *durum* × *Hordeum chilense*)	-	-	Tritordeum	A^u^BH	-	-	6×	CNR IBBA
*Triticum aestivum* ssp. *spelta* (L.) Thell	Test Altgold Rotkorn	ALT1	Spelt wheat	A^u^BD	PS	*T. spelta* Oberkulmer × *T. spelta* Sandmeier	6×	CREA-AA
	-	SPE			PS	-	6×	CNR IBBA
	Altgold Rotkorn	FAR111			PS	T. spelta Oberkulmer × T. spelta Sandmeier	6×	CREA-FLC
	Rossella	RO			CS	(Altgold rotkorn × Spada) × line Altgold	6×	CREA-AA
	Maddalena	MA			CS	T. spelta AltGold RotKorn × T. aestivum cv. Centauro	6×	CREA-AA
	Rita	RI			CS	T. spelta AltGold RotKorn × T. aestivum cv. Centauro	6×	CREA-AA
	Benedetto	BE			CS	T. spelta AltGold RotKorn × T. aestivum cv. Centauro	6×	CREA-AA
	Pietro	PI			CS	T. spelta AltGold RotKorn × T. aestivum cv. Spada	6×	CREA-AA
	Giuseppe	GI			CS	T. spelta AltGold RotKorn × T. aestivum cv. Bolero	6×	CREA-AA
	Montefortino’s Ecotype	FAR29			CS	-	6×	CREA-FLC
	-	FAR30			CS	-	6×	CREA-FLC
	Rubbiano’s Ecotype	FAR62			CS	-	6×	CREA-FLC
	Rouquin	FAR63			CS	(Lignée24 × Ardenne spelt) × Altgold	6×	CREA-FLC
	Impero	FAR106			CS	-	6×	CREA-FLC
*Triticum aestivum* ssp. *aestivum*	Abbondanza	FT4	Common wheat	A^u^BD	AE	-	6×	DISTAL
	Autonomia A	AUT A			AE	-	6×	DISTAL
	Autonomia B	AUT B			AE	-	6×	DISTAL
	Benco	BEN			AE	-	6×	DISTAL
	Bianco Nostrale	BNS			AE	-	6×	DISTAL
	Bilancia	BIL			AE	-	6×	DISTAL
	Bolero	FT5			AE	-	6×	DISTAL
	Carosello	FT7			AE	-	6×	DISTAL
	Eureka	FT8			AE	-	6x	DISTAL
	Frassineto	FT9			AE	-	6×	DISTAL
	Gentil Bianco	FT10			AE	-	6×	DISTAL
	Gentil Rosso Aristato	GRA			AE	-	6×	DISTAL
	Gentil Rosso Mutico	GRM			AE	-	6×	DISTAL
	Inallettabile	FT11			AE	-	6×	DISTAL
	Marzuolo D’Aqui	MAQ			AE	-	6×	DISTAL
	Mieti	MI			AE	-	6×	DISTAL
	Palesio	FT13			AE	-	6×	DISTAL
	Postarello	FT14			AE	-	6×	DISTAL
	San Francisco	FT15			AE	-	6×	DISTAL
	Sieve	SIE			AE	-	6×	DISTAL
	Terricchio	TRR			AE	-	6×	DISTAL
	Verna	FT17			AE	-	6×	DISTAL
*Zea mays* L.	Belgrano	-	Maize	-	-	-	-	CNR IBBA
*Oryza sativa* L.	Arborio	-	Barley	-	-	-	-	CNR IBBA
*Sorghum halepense* (L.) Pers.	-	-	Sorghum	-	-	-	-	CNR IBBA
*Avena sativa* L.	-	-	Oat	-	-	-	-	CNR IBBA
*Panicum miliaceum* L.	-	-	Millet	-	-	-	-	CNR IBBA
*Secale cereale* L.	-	-	Rye	-	-	-	-	CNR IBBA
*Hordeum vulgare* L.	-	-	Barley	-	-	-	-	CNR IBBA

Provider: DISTAL, Department of Agricultural and Food Sciences, Alma Mater Studiorum Università di Bologna. CNR IBBA, National Research Council—Institute of Agricultural Biology and Biotechnology; CREA-AA, Consiglio per la ricerca in agricoltura e l’analisi dell’economia agraria—Agricoltura e Ambiente, Headquarters of Foggia; CREA-FLC, Consiglio per la ricerca in agricoltura e l’analisi dell’economia agraria; Centro Ricerca Produzioni Foraggere e Lattiero Caseari; PS = pure spelt; CS = crossed. spelt; AE = soft wheat.

**Table 2 genes-13-00633-t002:** CE-*TUBB*7 numerical profile obtained via the amplification of target genome sequences present in 14 spelt and 22 common wheat accessions. Both the peak size (base pair) and height (RFUs—relative fluorescence units) of each profile are reported. On the right side, a comparison of the output obtained with the use of the three markers (*Q-locus*, *γ.gliadin-D* and *TUBB*7) reveals different spelt and common wheat discrimination success rates.

Type	ID	CE-*TUBB*7			*Q-Locus*	GAG65D	*TUBB*7	Type	ID	CE-*TUBB*7			*Q-Locus*	GAG65D	*TUBB*7
**pure spelt**	ALT1	Size	288		S	S	S	**soft wheat**	BNS	Size	288.5	300.9	W	W	W
		Height	31798							Height	16251	15643			
	SPE	Size	287.9		S	S	S		BIL	Size	288.3	300.8	W	W	W
		Height	32775							Height	32222	32237			
	FAR111	Size	288		S	S	S		FT5	Size	288.5	300.6	W	W	W
		Height	32280							Height	32205	32072			
**crossed spelt**	RO	Size	288.1	300.6	S	S	W		FT7	Size	287.8	300.1	W	S	W
		Height	31002	31350						Height	27895	27339			
	MA	Size	288.0	301.2	S	S	W		FT8	Size	288.6	300.7	W	W	W
		Height	32093	23482						Height	29673	32471			
	RI	Size	288.1		S	W	S		FT9	Size	288.2	300.5	W	W	W
		Height	32032							Height	31930	31540			
	BE	Size	288.1		S	S	S		FT10	Size	288.1	300.5	W	W	W
		Height	31759							Height	32225	31876			
	PI	Size	288		S	W	S		GRA	Size	288.5	300.8	W	S	W
		Height	31585							Height	30956	5438			
	GI	Size	288.2	300.7	S	W	W		GRM	Size	288.6	300.9	W	W	W
		Height	30719	31315						Height	26204	2526			
	FAR29	Size	287.9		S	S	S		FT11	Size	288.5	300.9	W	W	W
		Height	32191							Height	16332	15430			
	FAR30	Size	288.1		S	S	S		MAQ	Size	288.2		W	W	S
		Height	32360							Height	32238				
	FAR62	Size	288		S	S	S		MI	Size	288.5	301.1	W	S	W
		Height	31889							Height	25407	23928			
	FAR63	Size	288.1		S	S	S		FT13	Size	288.2		W	W	S
		Height	32178							Height	32323				
	FAR106	Size	287.7		S	S	S		FT14	Size	288.1	300.4	W	W	W
		Height	32243							Height	3965	31674			
**soft wheat**	FT4	Size	288.6	300.8	W	W	W		FT15	Size	288.2	300.5	W	W	W
		Height	27753	32535						Height	32076	31914			
	AUT A	Size	288.0	300.6	W	W	W		SIE	Size	288.6	300.8	W	W	W
		Height	30916	30903						Height	28759	30866			
	AUT B	Size	288.5	300.9	W	W	W		TRR	Size	288.5	300.9	W	W	W
		Height	16332	15430						Height	6193	5903			
	BEN	Size	288.1	300.6	W	S	W		FT17	Size	288.5	300.9	W	W	W
		Height	31306	30877						Height	5766	6412			

S = spelt genotype; W = wheat genotype.

**Table 3 genes-13-00633-t003:** List of the cereal-based flour samples tested using the *Q-locus* and TBP assays.

Code	Commercial Sample	Declared Composition
A	Spelt flour ‘Nobile’	Spelt wheat
B	Spelt flour ‘Bianca’	Spelt wheat
C	Wholemeal spelt flour ‘S’	Spelt wheat
D	Wholemeal spelt flour ‘V’	Spelt wheat
E	Semi-wholemeal spelt flour	Spelt wheat
F	White einkorn flour	Einkorn
G	Wholemeal einkorn flour	Einkorn
H	Wholemeal Khorasan Kamut^®^ flour	Khorasan Kamut^®^ wheat
I	Khorasan Kamut^®^ flour ‘type 0′	Khorasan Kamut^®^ wheat
L	Wholemeal emmer flour	Emmer
M	Wholemeal spelt flour ‘T’	Spelt wheat
N	Wholemeal multigrain flour	Spelt-soft wheat-durum wheat, einkorn, oat, barley, maize, rye

**Table 4 genes-13-00633-t004:** CE-TBP 1st intron numerical profiles obtained from the analysis of different cereal-based commercial flours and reference materials. Only the intron sizes are reported. Sample codes and compositions are the same as those shown in Table 3. Specific amplicons of undeclared ingredients (contaminations) are highlighted by different colored boxes. A color code is provided at the bottom.

**Food samples**	A	Spelt flour ‘Nobile’	Size	371	380	383	-	394	-	402	-	433	436	-	-	-	-	-	-	568	-	-	581	-	-	-	759	768	-	790	-	-	797	-	-	808	-	844	850	-	-	-	-	-	1151
	B	Spelt flour ‘Bianca’	Size	371	380	383	-	394	-	402	-	433	436	-	-	-	-	-	-	568	-	-	581	-	-	-	759	768	-	790	-	-	-	-	-	808	-	844	850	-	-	-	-	-	1151
	C	Wholemeal spelt flour ‘S’	Size	371	380	383	-	395	-	402	-	433	436	438	-	-	-	-	-	568	-	-	581	-	-	-	759	768	-	790	-	-	-	-	-	808	-	844	850	-	-	-	-	-	1152
	D	Wholemeal spelt flour ‘V’	Size	371	380	383	-	394	-	402	-	433	436	-	-	-	-	-	-	568	-	-	581	-	-	-	759	768	-	790	-	-	797	-	-	808	814	844	849	-	-	-	-	-	1151
	E	Semi-wholemeal spelt flour	Size	371	380	383	-	394	-	402	-	433	436	-	-	-	-	-	-	568	-	-	581	-	-	-	759	768	-	790	-	-	-	-	-	808	-	844	850	-	-	-	-	-	1152
	F	White einkorn flour	Size	371	380	383	-	394	-	402	-	433	436	438	-	-	-	-	-	-	570	-	581	-	-	-	759	768	-	790	-	-	-	-	800	808	814	844	850	-	-	-	-	-	1152
	G	Wholemeal einkorn flour	Size	371	380	383	-	394	-	402	-	-	436	-	-	-	-	-	-	-	570	-	581	-	-	-	759	-	-		-	-	-	-	800	-	814	844	850	-	-	-	-	-	1152
	H	Wholemeal Khorasan Kamut® flour	Size	371	380	383	-	394	-	402	-	433	436	-	-	-	-	-	-	568	-	-	581	-	-	-	759	768	-	790	-	-	797	-	-	808	-	844	850	-	-	-	-	-	1152
	I	Khorasan Kamut® flour ‘type 0’	Size	371	380	383	-	394	-	402	-	433	436	-	-	-	-	-	-	568	-	-	581	-	-	-	759	768	-	790	-	-	797	-	-	808	-	844	850	-	-	-	-	-	1152
	L	Wholemeal emmer flour	Size	371	380	383	-	394	-	402	-	433	436	-	-	-	-	-	-	568	-	-	581	-	-	-	759	768	-	790	-	-	797	-	-	808	-	844	850	-	-	-	-	-	1152
	M	Wholemeal spelt flour ‘T’	Size	371	380	383	-	394	-	402	-	433	436	-	-	-	-	-	-	568	-	-	-	-	-	-	759	768	-	790	-	-	-	-	-	808	-	844	849	-	-	-	-	-	1150
	N	Wholemeal multigrain flour	Size	371	380	383	390	394	398	402	421	433	436	-	483	503	515	520	566	568	-	574	581	593	604	749	759	768	772	790	792	796	797	799	800	808	-	844	849	871	901	999	1008	1024	1150
**Reference materials**	-	Spelt seed	Size	371	380	383	-	394	-	402	-	433	436	-	-	-	-	-	-	568	-	-	-	-	-	-	759	768	-	790	-	-	-	-	-	808	-	844	850	-	-	-	-	-	1151
	-	Common wheat seed	Size	371	380	383	-	394	-	402	-	433	436	438	-	-	-	-	-	568	-	-	581	-	-	-	759	768	-	790	-	-	-	-	800	-	-	844	850	-	-	-	-	-	1151
	-	Durum wheat seed	Size	371	380	383	-	394	-	402	422	433	436	-	-	-	-	-	-	-	-	-	581	-	-	-	759	-	-	-	-	-	797	-	-	808	-	844	-	-	-	-	-	-	1149
	-	Einkorn seed	Size	371	380	-	-	394	-	-	-	-	436	-	-	-	-	-	-	-	570	-	-	-	-	-	759	-	-	-	-	-	-	-	-	-	814	-	-	-	-	-	-	-	-
	-	Emmer seed	Size	371	380	383	-	394	-	402	422	433	436	-	-	-	-	-	-	-	-	-	581	-	-	-	759	768	-	-	-	-	-	-	-	808	-	844	-	-	-	-	-	-	1149
	-	Khorasan seed	Size	371	380	383	-	394		402	422	433	436	-	-	-	-	-	-	-	-	-	581	-	-	-	759	-	-	-	-	-	797	-	-	808	-	844	-	-	-	-	-	-	1149
	-	Maize seed	Size	-	-	-	-	-	398	-	-	-	-	-	-	-	-	-	566	-	-	-	-	593	604	-	-	-	-	-	-	-	-	-	-	-	-	844	-	-	901	1000	1009	-	-
	-	Oat seed	Size	-	-	-	390	-	-	-	-	-	-	-	483	503	515	520	-	-	-	-	-	-	-	749	-	-	-	-	-	-	-	-	800	-	-	-	-	872	-	-	1009	1024	-
	-	Rye ssed	Size	-	-	-	-	394	-	-	420	-	-	-	-	-	-	-	-	-	-	-	-	-	-	-	-	-	-	-	792	796	-	-	-	-	-	-	-	-	-	-	-	-	-
	-	Barley seed	Size	-	-	-	-	394	-	-	421	-	-	-	-	-	-	-	-	-	-	574	-	-	-	-	-	-	773	-	-	-	-	-	801	-	-	-	-	-	-	-	-	-	-

*T. monococcum ssp. monoccocum* (einkorn) 

, Durum wheats (durum or Kourasan wheat) 

, Hexaploid wheats (common or spelt weat) 

.

## Data Availability

Data are contained within the article or Appendix A.

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
