# Peer review of "A Combinatorial *Q-Locus* and Tubulin-Based Polymorphism (TBP) Approach Helps in Discriminating *Triticum* Species"

_genes, 2022, doi:10.3390/genes13040633_

Round 1

Reviewer 1 Report

The article "A Combinatorial Q-locus and Tubulin Based Polymorphism (TBP) Approach Helps in Discriminating Triticum Species" is a well written article and provides an in-depth discernment about differentiating between the different species of Triticum. There are few comments which I have made (File attached) that if valid, needs to be addressed. Overall, the article is of good significance to people involved in wheat flour business and its supplies to the industry. In addition, since the method described is straightforward, it can be used by bakery staff to differentiate the wheat flour if the need arises.

Good Luck! 

Author Response

On behalf of all the authors I wish to thank the referees for their kind and generous evaluation of our work. In this revised version of the paper, we have  introduced all the changes requested. We are sorry we could not adequately answer to the criticism of  some lack of clarity made by ref 2 but in the absence of any specific remark, and in presence of two contrasting judgments, we could not do any better. We are confident though that a higher clarity may now have been achieved  at least in the introduction section where we modified several sentences to lower the duplication rate, as recommended by the handling Editor. Best regards and thanks again. Diego Breviario

Reviewer 2 Report

The MS investigates the best approach to distinguish spelt from common wheat. The authors claim that all the Triticum species can be precisely identified utilizing the dual approach of flanking the Q-locus assay with TBP-analysis. While this was a laudable goal, the MS suffers from clarity.

Author Response

(The authors gave the same response as above.)

Reviewer 3 Report

In this study, authors compared three previously published methods (Q-locus and γ-gliadin encoding locus, and TBP (Tubulin Based-Polymorphism)) for the identification of genetic origin of wheat species. They applied all three methods on different wheat species and commercial wheat-based flours and discussed their suitability for the best detection. This study is well planned and offer the possibilities to adopt the combination of methods for commercial settings. The article is well written: introduction, and material and methods are properly described; results are nicely explained and discussed. However, authors should take care of small typo errors, such as γ-gliadin (line 301, 384 etc).

Author Response

(The authors gave the same response as above.)
